# Correlates of participation in community-based interventions: Evidence from a parenting program in rural China

**Yiwei Qian** [1]*, **Yi Ming Zheng**[2], **Sarah-Eve Dill**[2], **Scott Rozelle**[2]

**1** Department of Economics, University of Southern California, Los Angeles, California, United States of America, **2** Rural Education Action Program, Stanford University, Stanford, California, United States of America

* yiweiqia@usc.edu

**Data Availability Statement:** Data cannot be shared publicly because the data contains sensitive information on the cognitive development of

## Abstract

A growing body of literature has documented that community-based early childhood development (ECD) interventions can improve child developmental outcomes in vulnerable communities. One critical element of effective community-based programs is consistent program participation. However, little is known about participation in community-based ECD interventions or factors that may affect participation. This paper examines factors linked to program participation within a community-based ECD program serving 819 infants and their caregivers in 50 rural villages in northwestern China. The results find that more than half of families did not regularly attend the ECD program. Both village-level social ties within the program and proximity to the program significantly predict program participation. Increased distance from the program site is linked with decreased individual program participation, while the number of social ties is positively correlated with participation. The average program participation rates among a family's social ties is also positively correlated with individual participation, indicating strong peer effects. Taken together, our findings suggest that attention should be given to promoting social interactions and reducing geographic barriers among households in order to raise participation in community-based ECD programs.

## Introduction

In low- and middle-income countries (LMICs), over 250 million children under 5 years old are at risk of not reaching their full developmental potential due to poverty and under-stimulation [1]. Such widespread developmental delays carry potential long-term consequences, as delays in early childhood have been shown to negatively impact educational attainment and income and perpetuate an intergenerational cycle of poverty [2, 3]. In response to these concerns, a growing body of literature has documented that parenting interventions, which aim to increase psychosocial stimulation in early childhood by training caregivers in age-appropriate interactive parenting practices [4], can improve early childhood development (ECD) in vulnerable communities [2, 5–7]. Studies of ECD parenting programs have found positive effects

infants in rural China. Data from the study are available upon request from Sean Sylvia, School of Public Health, University of North Carolina (contact via sean_sylvia@unc.edu) for researchers who meet the criteria for access to confidential data.

**Funding:** The funding sources that supported the study are the International Initiative for Impact Evaluation (3ie) and the 111 Project (Grant no: B16031). Neither of the funding sources for the present source are commercial sources. None of the funders had any role in study design, data collection and analysis, decision to publish, or preparation of the manuscript.

**Competing interests:** The authors have declared that no competing interests exist.

on cognitive and non-cognitive development in a number of developing settings, including Jamaica, Colombia, Pakistan, and China [2, 6, 8, 9].

While there are different approaches to training parents to provide a more stimulating environment for their young children, community-based ECD interventions have become a promising model for future ECD programs [10]. Community-based ECD programs provide parenting training and environmental stimulation through one-on-one and/or group activities based in a central location [11]. In addition to reducing labor and other costs associated with home visiting-based ECD interventions [10], community-based programs have been shown to effectively raise child developmental outcomes [12, 13]. Studies of community based ECD programs have returned positive impacts on child cognitive and non-cognitive skills development in Bangladesh, Brazil, Mozambique and Indonesia [11, 14–16].

Although community-based programs have been promoted as a new model for ECD interventions, low rates of participation can significantly hinder community-based programs from functioning effectively. While no studies have examined participation in community-based ECD programs specifically, randomized controlled studies of other (non-ECD) community-based interventions have found significant variation in impacts by individual participation [17, 18]. Low rates of participation can also limit the overall effects of an intervention: in a study of a community-based disease prevention program, O'Loughlin et al. (1999) [19] found that consistently low participation diluted the would-be program benefits, resulting in limited overall program effectiveness. In the case of ECD programs, low participation rates might mean that interventions would not have a community-wide impact on child development—even if the intervention is effective for those that do participate. In fact, in two studies of community-based ECD programs in Mexico and Colombia, researchers hypothesized that low rates of participation may have reduced the positive treatment effects on child skills development [20, 21].

Given the importance of participation for the success of community-based ECD programs, it is necessary to understand the factors that may limit participation in community-based programs. In the literature on program uptake, there are two potential factors that may impact participation. The first factor that may affect participation in community-based ECD programs is social ties within the community. That interactions within a social network can affect the decision making of prospective participants is well documented in the literature [22, 23]. For example, studies have found that social ties with peers can influence an individual's decision to use welfare and other social services [24, 25]. In addition, an individual's social capital within a community (sometimes defined as the number of social ties [26]) has also been found to contribute to participation in non-ECD social programs [27, 28]. Since the nature of community-based ECD programs involves reaching individuals with the existing social network of a community, social ties between families may affect whether and how individuals participate in a program. However, the role that social interactions play in participation in community-based ECD programs has not yet been studied.

The second factor is geographic proximity. In studies of non-ECD community-based programs, the literature suggests two possibilities regarding the relationship of geographic proximity to program participation. On the one hand, closer proximity to the program location should lead to easier access and lower costs of participation, which may lead nearby households to increase their participation [29]. On the other hand, the costs of increased distance may be outweighed by the potential returns to child development [30]. As with social ties, however, no study to date has examined the role of geography in participation in community-based ECD programs.

To address these gaps in the literature, this paper examines factors linked to program participation within a community-based ECD program in rural China, focusing specifically on

the roles of social ties and geographic proximity. To meet this goal, we have three specific objectives. First, we describe the overall participation in the community-based ECD program, as well as the geographic proximity of participants to the program sites and the social ties between participants. Second, we examine the associations between social ties and ECD program participation. Third, we analyze the correlations between geographic proximity and program participation.

In pursuit of these objectives, we draw on data from a community-based ECD program in rural China. The program built parenting centers in 50 rural villages located in northwestern China, serving a total of 819 infants age 6–24 months and their families. We collected detailed participation data for all households in the program, as well as the village-level social network information for each participant, geographic information on the distance from the parenting center to the home of each participant, and child and household characteristics of each participating family. Using this information, we evaluate the links between social interactions, geographic proximity and program participation while controlling for other household characteristics that may affect program participation.

Rural China can serve as a useful context to study these research questions for several reasons. First, China is representative of other LMICs in terms of its levels and dispersion of economic and social development. Economically it is middle income, but the country also has a large rural population that has not developed at the same speed as urban areas [31, 32]. Additionally, recent studies have found low levels of ECD in rural China, a common challenge to human capital accumulation across LMICs. A 2018 study of four major rural subpopulations in China found that 49% of children were cognitively delayed [33], a higher rate than the 43% of children at risk of long-term reduced cognition across all LMICs [34]. Since rural China shares the same problem of poor ECD outcomes as other LMICs, effective interventions in rural China can serve as a guideline for ECD interventions in similar contexts. Finally, evidence from this intervention can inform scalable ECD policies in China and other LMICs. Although the Chinese government has put forward a new ECD policy for its large rural population [35], policymakers are still debating the details of effective implementation. Evidence from our study can inform the implementation of this policy to maximize participation in ECD programs and improve child development.

The rest of this paper proceeds as follows. The next section describes our methods, including sample selection, data collection and statistical approach. Section III presents the results. Section IV discusses the results and concludes.

## Methods

### Sampling

Ethical approval for this study was granted by the Stanford University Institutional Review Board (IRB) (Protocol ID 35921). We draw our data from a community-based ECD intervention implemented in 20 nationally-designated poverty counties in the Qinling mountain region of northwestern China. To select the sample for the interventional study, the research team followed a three-step sampling protocol. First, all townships (the middle level of administration between county and village) in the 20 counties were selected to participate in the study, with two exceptions: the study excluded the township in each county that housed the county seat (as these tend to be wealthier and more urban than the average rural township), as well as townships that did not have any villages with a population of 800 or more. After applying the two exclusion criteria, the sample consisted of 100 townships.

Second, one village per township was randomly selected for inclusion in the study. To ensure that all sample villages would have the potential space to conduct the community-based

parenting intervention, villages that could not supply a 60–80 m$^2$ space for the intervention site were excluded. If a village did not have the available space, it was replaced with a randomly-selected village from within the same township. In total, 100 villages were included in our sample.

Finally, the research team selected sample families (i.e., children and their caregivers) for participation in the study. A list of all registered births over the past 24 months was obtained from the local family planning official in each sample village. All children in the desired age range (6–24 months) and their caregivers were enrolled in the interventional study. At baseline, the total sample included 1,664 children and their caregivers from 100 villages in the 20 sample counties.

After sampling, the research team randomly assigned 50 sample villages to the treatment arm and 50 villages to the control arm of the study (see Fig 1). In total, 794 children and their caregivers were assigned to the control arm of the study, and 870 children and their caregivers were assigned to the treatment arm. Parenting centers were built in all of the treatment villages, while the control villages did not receive any intervention. To meet the goal of this study —examining the program participation of community-based ECD interventions—our final sample includes the 870 children and caregivers in the 50 villages that comprised the treatment arm of the interventional study.

Our sample also excludes all long-term migrant households that did not raise their child in the village during the first year of operation of the local parenting center. Since households that had moved out of the village on a long-term basis would not feasibly have participated in the parenting centers, we exclude them from the sample to avoid underestimating the true rate of participation among households who were able to participate. In our sample, 51 households had out-migrated on a long-term basis. After excluding these 51 households, the final study sample includes 819 children aged 6–24 months and their caregivers from 50 sample villages. On overview of the sampling strategy is presented in Fig 1.

## Intervention

In each village, a parenting center was built by the research team using space provided by the local village committee. All parenting centers included a large play area, as well as toys, baby books and decorations provided by the research team. The parenting centers were designed to be open 5 hours a day and 6 days a week, and 90% of parenting centers were open for the designated frequency and time. On average, the parenting centers were open for 279 days during the first year of operation. Caregivers were encouraged to bring their children to the parenting

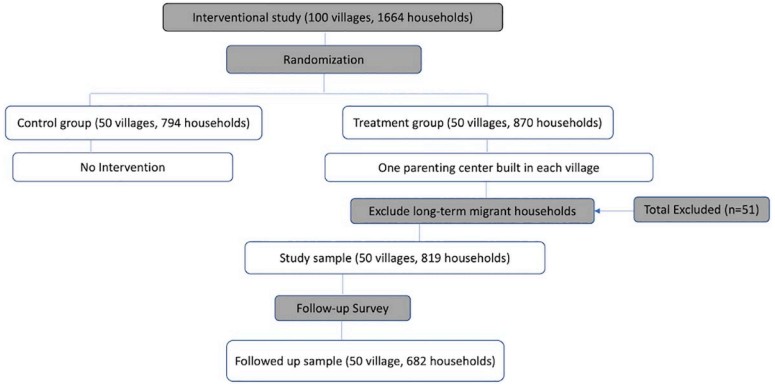

**Fig 1. Overview of sample selection.**

centers during open hours, but they were not allowed to leave their children alone in the parenting centers.

Each parenting center was staffed by two trained parenting experts from the local Family Planning Commission, who conducted weekly one-to-one parenting lessons for caregivers on interactive parenting practices to stimulate child development. The research team also hired one local center manager for each parenting center who was responsible for managing all center activities, including organizing daily group activities (such as storytelling and singing) and recording program participation. For each visit to the parenting center, the center manager recorded the caregiver's name, the date, and the relationship between the caregiver and the child.

## Data collection

The data presented in this study were collected in two survey rounds by teams of trained enumerators. The baseline survey, conducted between November 2015 and February 2017, collected data on child and household characteristics. In August 2018, we conducted a follow-up survey in which we collected information from sample households on program participation, social ties to other households within the program, and geographic proximity to the parenting centers.

**Baseline survey.**   The baseline survey collected socioeconomic and demographic data from all households participating in the study. For each household, the child's primary caregiver was identified as the person most responsible for the child's daily care (typically the child's mother or grandmother) and administered a detailed survey of child and household characteristics. Child characteristics include the child's gender, age in months, and whether the child has siblings. The exact age of each child was obtained from his or her birth certificate. Household characteristics include identity of the primary caregiver (parent, grandparent, or other), the primary caregiver's education level, whether the primary caregiver spends time on nonfarm work, whether the father has out-migrated for work, and household wealth. To measure household wealth, we created a household asset index using polychoric principal component analysis based on whether the household owned the following items: tap water, a toilet, a water heater, a computer, Internet access, a refrigerator, an air conditioner, a motorcycle or electric bicycle, and a car.

We also measured child cognitive development using the cognitive scale of the Bayley Scales of Infant Development-Third Edition (BSID-III) [36]. The BSID-III was administered in the home of each sample child using a standardized set of toys and a detailed scoring sheet. We compute age-adjusted internal z-scores from the raw scores for cognitive development by subtracting the age-specific means and dividing by the age-specific standard deviations estimated using non-parametric regression methods [37]. Cognitive delay is defined as cognitive scale scores one or more standard deviations (SD) below the mean of a healthy population, which studies have shown to be 105 with SD of 9.6 [38, 39]. We use the BSID-III age-standardized cognitive scores as a control variable in our subsequent analysis.

**Participation records.**   Throughout the one-year intervention, we collected parenting center participation records for all sample children and their caregivers. During the intervention, the parenting center managers recorded each visit to the center, including the child's name, the date, and the relationship between the caregiver and the child. The research team then collected the participation records from the center manager at each parenting center. We calculated a participation rate for each child as the ratio of days that the child and caregiver visited the parenting center to the total number of open days.

**Follow-up survey.**   The follow-up survey collected two blocks of data. First, we collected information about social interactions between households in each sample village. Since we are

interested in the social interactions between households eligible to participate in the ECD program, we limited our survey to the social network of eligible households in each sample village. We first compiled a roster of program participants for each village, and then used the roster to ask households to identify the households with whom they interacted and how often they interacted. Interactions were not limited exclusively to those between heads of households or topics about children or parenting. Collecting data on the full range of social interactions between households reduces the chance of measurement error and is commonly used to assess social networks in the literature [40].

Fig 2 shows the social network structure of households eligible to attend the parenting centers within a randomly chosen village in our sample. In this study, households that interact at least once per week are considered to have a social tie. Using this information, we generated three variables for each household: the number of social ties, the average distance of social ties to the parenting center, and average participation rate of social ties in the parenting center.

In addition, we measured the exact travel distance and route to the parenting center for each sample household during the follow-up survey. To do so, enumerators first located the waypoint (measured by longitude and latitude) of each sample household using the Global

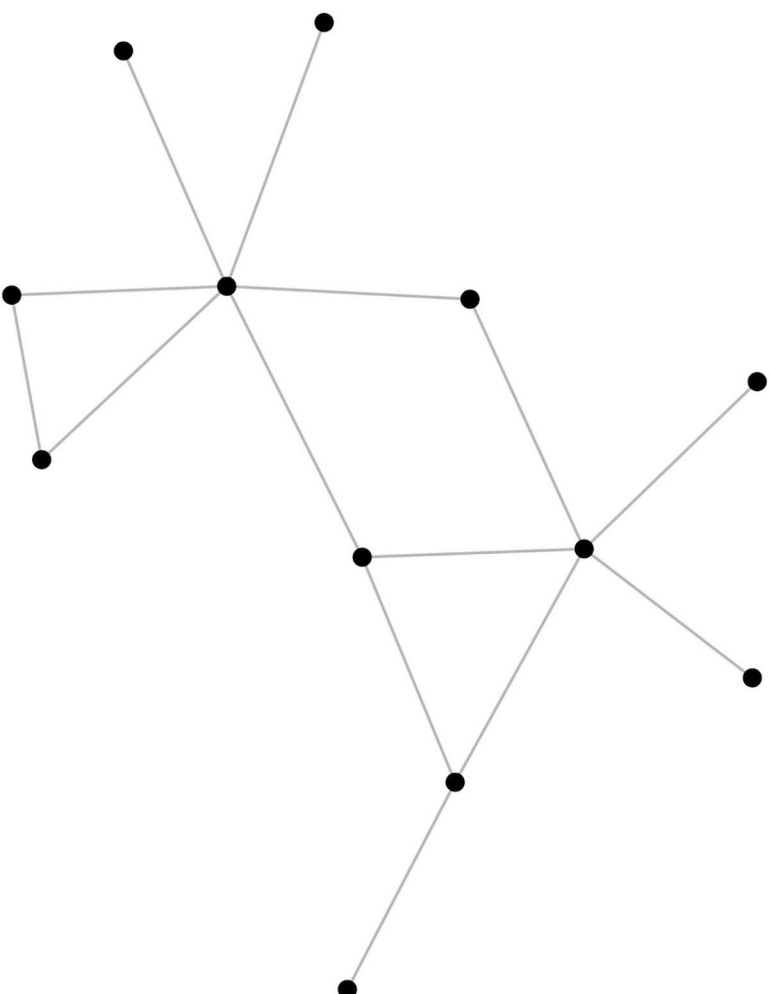

**Fig 2. Social network structure of households in one sample village (N = 12).** Each node represents one household. Connecting lines indicate social ties between households (defined as interacting at least one time in the past week).

Positioning System (GPS) available through the Motion X-GPS smartphone application (V24.2, Fullpower Labs, Santa Cruz, CA). Enumerators then followed the caregiver's typical route to the parenting center while recording the GPS track of the route. Both waypoints and GPS tracks were imported to the Geographic Information System (GIS) using the software QGIS (V3.6, OSGeo, Beaverton, OR) to calculate the distance between each sample household and its corresponding parenting center.

## Statistical analysis

All statistical analyses were performed using STATA 14.0. P-values below 0.05 were considered statistically significant. STATA's multiple linear regression model was used to conduct the multivariate analysis. We included the following variables as potential confounders in the multivariate analysis: child age and gender, whether the child has siblings, baseline cognitive development (BSID-III age-standardized score), whether either grandparent is the primary caregiver, educational level of the primary caregiver, whether the primary caregiver has nonfarm work, whether the father has out-migrated, and household asset index (polychoric principal component score). To account for the nested nature of the data, we cluster all standard errors at the village level.

## Results

### Summary statistics

Table 1 reports the socioeconomic characteristics of children and households in the study sample. At the time of the baseline survey, 51.4% of the sample children were male, and 49.2% had no siblings. The data also show that 54.3% of the sample children were cognitively delayed at baseline (defined as BSID-III cognitive scale scores one or more SD below the healthy mean). For 28.7% of sample children, a grandparent was identified as the primary caregiver, while 69.2% of children were primarily cared for by the mother. Only 7.2% of primary caregivers engaged in nonfarm work. A total of 15.4% of primary caregivers in our sample had completed

**Table 1. Summary statistics.**

| Variables | Mean | SD | Observations |
|---|---|---|---|
| **Child characteristics** | | | |
| Age (months) | 14.356 | [5.42] | 814 |
| Gender (1 = male) | 0.514 | [0.500] | 819 |
| Only child (1 = yes) | 0.492 | [0.500] | 819 |
| Cognitive delay (BSID-III cognitive scale score < -1 SD) | 0.543 | [0.498] | 814 |
| **Caregiver characteristics** | | | |
| Grandparent is primary caregiver (1 = yes) | 0.287 | [0.453] | 819 |
| Mother is primary caregiver (1 = yes) | 0.692 | [0.462] | 819 |
| Other relative is primary caregiver (1 = yes) | 0.021 | [0.143] | 819 |
| Primary caregiver spends time on nonfarm work (1 = yes) | 0.072 | [0.258] | 819 |
| Primary caregiver education (1 = at least 9 years) | 0.154 | [0.361] | 819 |
| **Family characteristics** | | | |
| Household asset index | 0.145 | [1.197] | 819 |
| Father has out-migrated (1 = yes, 0 = no) | 0.527 | [0.500] | 819 |

Distances from the parenting center were measured at the follow-up survey for the non-attrited sample.

Data source: Authors' survey.

**Table 2. Participation rates in community-based ECD program.**

| Panel A: Overall participation rate | | | | |
|---|---|---|---|---|
| | | Mean | SD | Obs |
| Visits to parenting center of total open days | | 0.336 | [0.293] | 682 |
| Panel B: Participation by caregiver type | | | | |
| | | Percentage | | |
| | Share of visits accompanied by parents | 53.57% | | |
| | Share of visits accompanied by grandparents | 44.64% | | |
| | Share of visits accompanied by other relatives | 1.78% | | |

nine years of schooling or more, and over half (52.7%) of fathers in our sample had out-migrated from the village for work.

## Participation in the parenting centers

Table 2 presents the overall participation rate for the full sample, as well as participation by primary caregiver type (parents, grandparents, or other relatives). The overall participation rate was 33.6%, meaning that sample households took their children to the parenting centers for one third of open days, or about two days per week on average. Parents accompanied children on 54% of visits, while grandparents accounted for 45% of visits and other relatives accounted for the remaining 1%.

Fig 3 shows the breakdown of participation rates within in the parenting centers. The data show that 50.4% of sample households attended fewer than 20% of open days, or less than

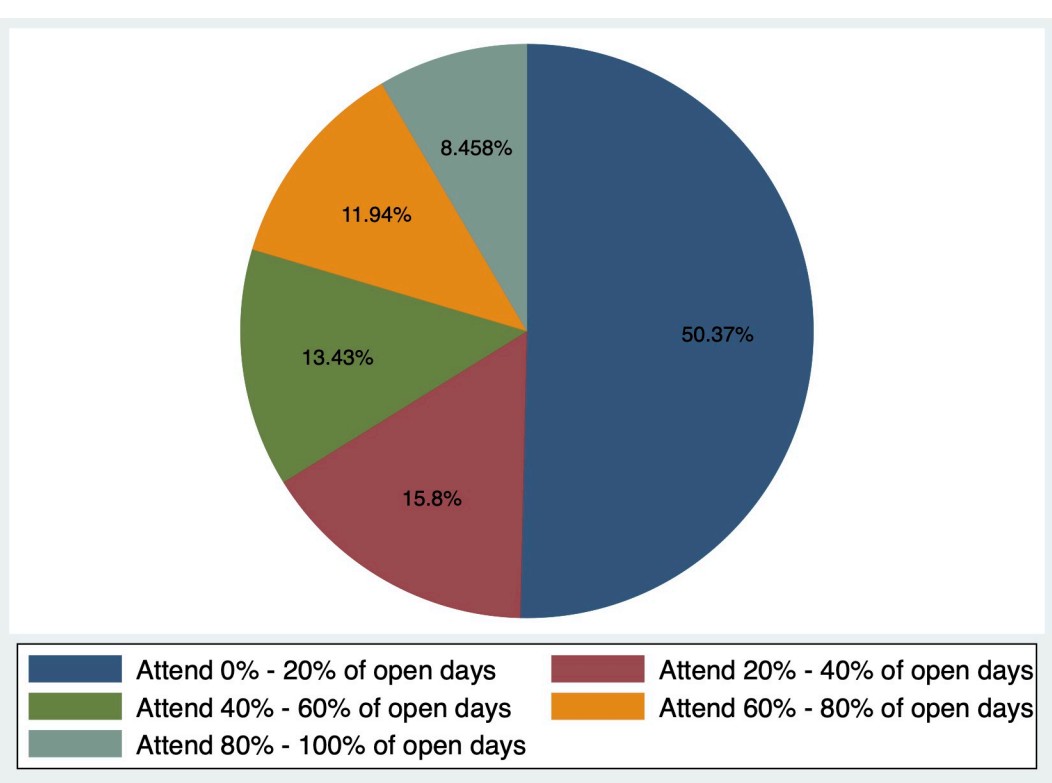

**Fig 3. Parenting center participation rates.** The total number of open days is 168–316 across 50 parenting centers, with an average of 279 days open. Of the 50 parenting centers, 90% were open between 246 and 311 days in total. Participation data was collected over one year.

once per week on average. Only 8% of households attended more than 80% of open days (4 to 5 days per week).

## Social interactions between households and geographic proximity to the program

Table 3 presents the social network information of households in the study sample. On average, each household had social ties with five other households in the program, and 50.4% of households had four social ties or fewer (Table 3 Panel B). The average distance of a household's social ties to the parenting center was 0.86 kilometers. As shown in Panel C, more than 40% of households had friends who lived within 0.5 km of the parenting center. Finally, the average participation rate (SD) of a household's social ties was 47% (36.7).

Table 4 presents the geographic proximity of sample households to the program. As seen in Panel A, households in our sample lived an average of 0.97 kilometers (km) from their local parenting center. Panel B shows the distribution of households by distance from the parenting centers. A large share (47%) of households lived within 0.5 km of their local parenting center, and another 22% of households lived between 0.5 km and one km from the parenting center. Only 12% of households were more than two km away from the parenting center.

## Correlates of participation

The main objective of this paper is to examine whether social network characteristics and geographic proximity are associated with participation in the community-based ECD program. To meet this objective, we first examine the bivariate correlation between social ties and program participation. Figs 4, 5, and S1 Fig present the correlations between program participation and the number of social ties, average participation rate of social ties, and average distance of social ties to the program, respectively. As we can see, the number of social ties is

**Table 3. Summary statistics of social interactions of non-attrited households.**

| Panel A: Social ties between eligible households | | | |
|---|---|---|---|
| Variable Name | Mean | SD | Obs |
| Number of social ties | 5.021 | [3.493] | 682 |
| Average distance of social ties to parenting center (km) | 0.856 | [0.944] | 682 |
| Average participation rate of social ties | 0.472 | [0.367] | 682 |
| Panel B: Distribution of number of social ties | | | |
| | | Percentage | |
| | 0–2 friends | 26.25% | |
| | 3–4 friends | 24.19% | |
| | 5–6 friends | 22.43% | |
| | ≥ 7 friends | 27.13% | |
| Panel C: Distribution of distance to the program among social ties | | | |
| | | Percentage | |
| | 0–0.5 km | 43.11% | |
| | 0.5–1 km | 29.77% | |
| | 1–1.5 km | 12.90% | |
| | 1.5–2 km | 6.30% | |
| | > 2 km | 7.92% | |

Source: Authors' survey.

**Table 4. Summary statistics for geographic proximity and social ties of non-attrited households.**

| Panel A: Distance of sample households to the program | | | | |
| --- | --- | --- | --- | --- |
| | | Mean | SD | Obs |
| Distance to the program (km) | | 0.965 | [1.201] | 675 |
| Panel B: Distribution of distance to the program | | | | |
| | | Percentage | | |
| | 0–0.5 km | 46.67% | | |
| | 0.5–1 km | 21.93% | | |
| | 1–1.5 km | 12.89% | | |
| | 1.5–2 km | 5.63% | | |
| | > 2 km | 12.89% | | |

Source: Authors' survey.

significantly and positively correlated with higher participation in the parenting centers (Fig 4). For household with 0–2 social ties, the participation rate was 23.8%, while households with 3–4 social ties, 5–6 social ties, and seven or more social ties had participation rates of 29.7%, 38.3% and 44.5%, respectively. The rate of participation among a household's social ties is also positively correlated with higher individual participation (Fig 5). However, we do not find a consistent bivariate correlation between the average distance of social ties to the program and program participation (S1 Fig).

We also examine the bivariate relationship between geographic proximity and program participation. The results, presented in Fig 6, show a clear correlation between proximity and

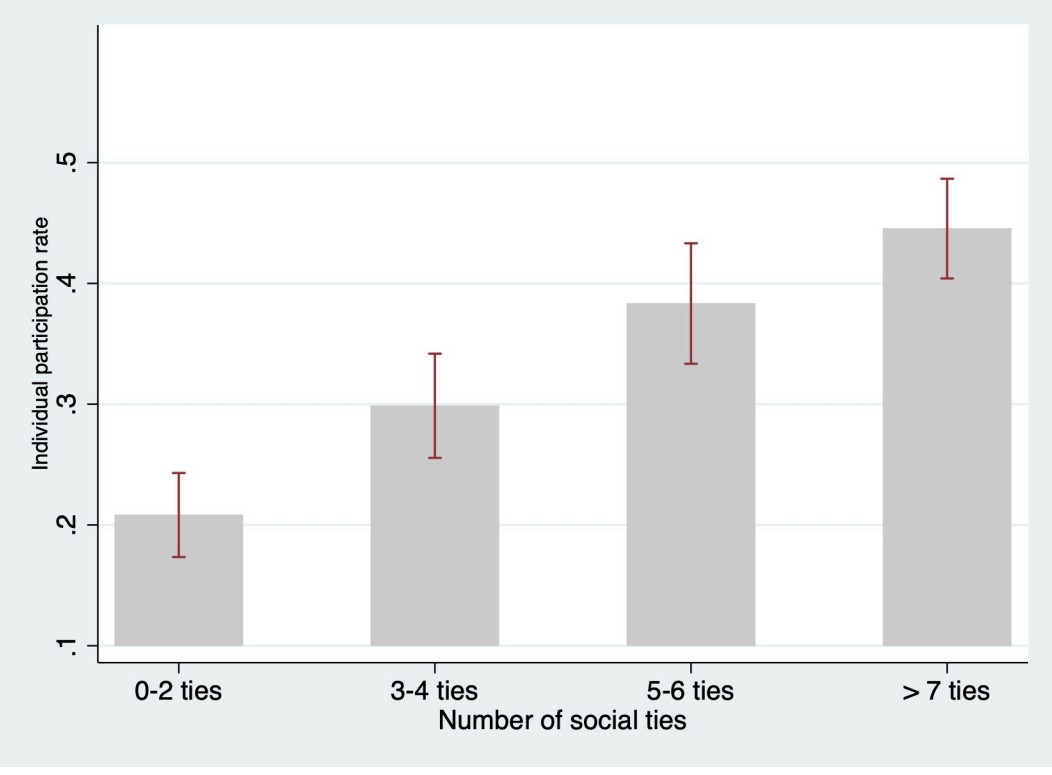

**Fig 4. Participation rates and social ties of households.** Source: Authors' survey.

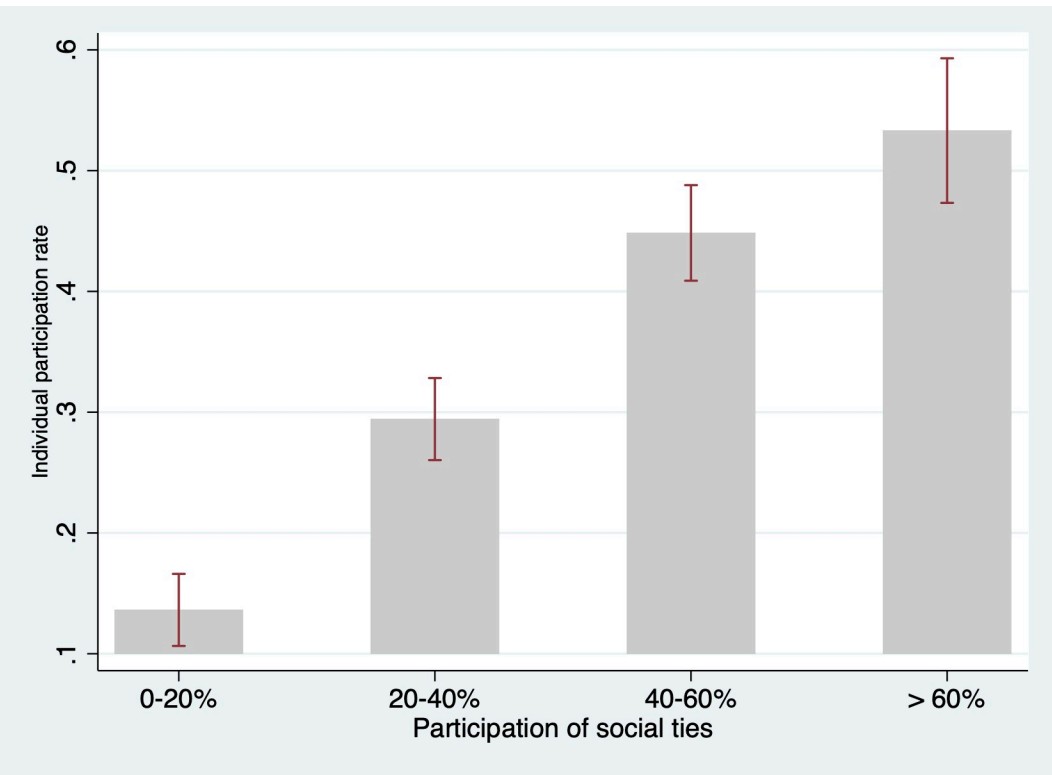

**Fig 5. Participation rates and the participation of social ties.** Source: Authors' survey.

program participation, with households closer to the parenting centers showing significantly higher participation rates. Specifically, households living within 0.5 km of the parenting center attended 45% of open days, while the participation rate decreased to 32% of open days for households living 0.5 km to 1 km from the parenting center. This difference is significant at $p<0.05$ as the 95% confidence interval does not overlap. For households living 1 km to 1.5 km and 1.5 km to 2km from the parenting centers, the participation rate further decreased to 22% and 10% of open days, respectively (both $p<0.05$).

Finally, we examine the correlations between geographic proximity, social interactions and participation in the parenting center program after controlling for various confounders. To do so, Table 5 presents the result of our multivariate regression holding constant all baseline child and household characteristics. We find that both the number of social ties and the participation rate of social ties are significantly and positively associated with increased participation in the parenting centers. Specifically, one additional social tie is associated with a 2.4-percentage-point increase in attendance at the parenting centers ($p<0.001$). Additionally, for each 10-percentage-point increase in the average participation rate of a household's social ties, we find a 2.2-percentage-point increase in that household's own participation rate ($p<0.05$). However, we do not find a significant relationship between the average distance of a household's social ties from the parenting center and program participation.

In addition, consistent with the results of Fig 6, the data show a negative relationship between geographic proximity and participation rates. In column 1 (when social network characteristics are not controlled for) a one-km increase in distance corresponds to a decrease in participation by about nine percentage points ($p<0.001$). After adding social network characteristics (columns 2, 3, & 4), geographic proximity is still negatively correlated with

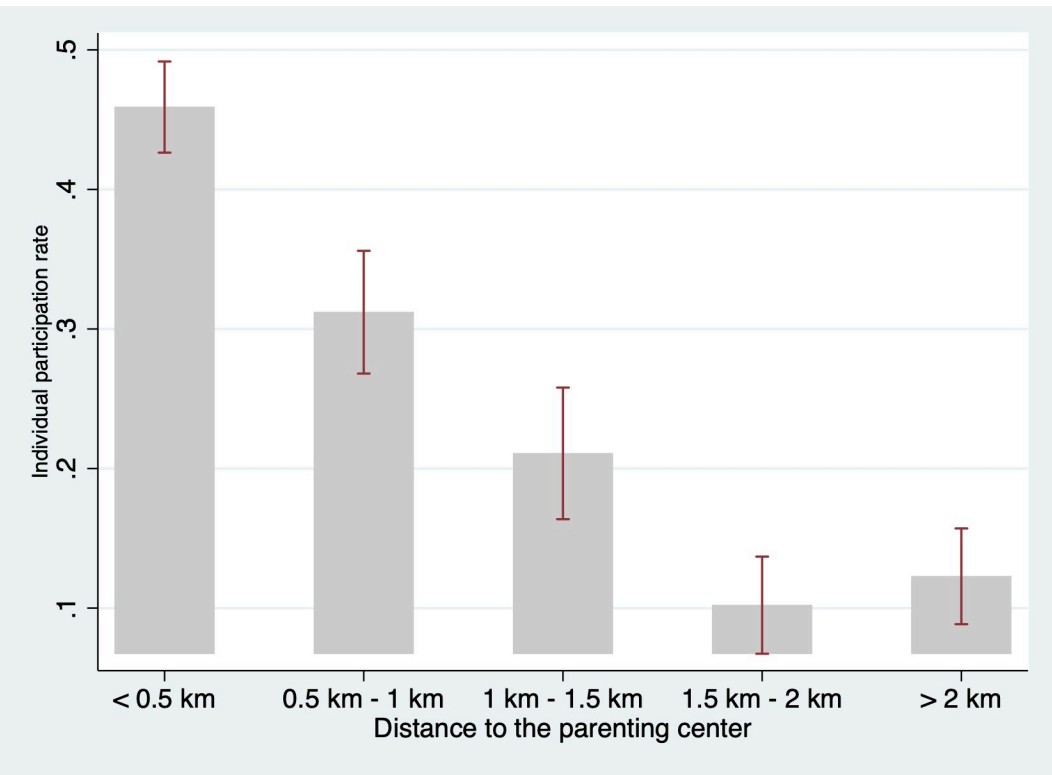

**Fig 6. Participation rates and distance to the parenting center.** Source: Authors' survey.

participation. In our most preferred specification (column 4), which controls for all social network characteristics, each one-km increase in distance from the parenting center corresponds to a decrease in participation by about seven percentage points (p<0.01).

To check the robustness of our findings, we performed the same multivariate regression analysis as Table 5, using two different definitions social ties. S2 Table shows the distribution of number of social ties in our sample, using two alternative definitions for social ties. Panel A shows the distribution when social ties are defined as interacting at least once per month (less frequent than the original definition), and Panel B presents the distribution when social ties are defined as interacting at least twice per week (more frequent than the original definition). S3 Table and S4 Table present the regression results with social ties defined as interacting at least once per month and defined as interacting at least twice per week, respectively. The results find consistent correlations between the number of social ties and program participation, regardless of the definition of a social tie. However, although correlation between the average participation rate of social ties and individual participation remains significant when we use the stricter definition of a social tie (at least twice a week; S4 Table), the correlation is not statistically significant when we use the looser definition of social ties (at least once a month; S3 Table). Geographic proximity remains significantly correlated to program participation at a similar magnitude in all tables.

## Discussion

Ensuring sufficient participation is essential to the effectiveness of community-based programs. This paper examines factors associated with participation in a community-based ECD intervention in rural China, focusing specifically on the roles of geographic proximity and

Table 5. Correlates of participation in the community-based ECD program.

| | (1) | (2) | (3) | (4) |
|---|---|---|---|---|
| | | Participation rate | | |
| Number of social ties | | 0.025*** | 0.025*** | 0.024*** |
| | | (0.006) | (0.006) | (0.006) |
| Average distance of social ties to program | | | -0.004 | 0.002 |
| | | | (0.017) | (0.014) |
| Average participation of social ties | | | | 0.222* |
| | | | | (0.103) |
| Distance to the program (km) | -0.091*** | -0.075** | -0.074** | -0.070** |
| | (0.022) | (0.024) | (0.026) | (0.026) |
| Male child | 0.009 | 0.009 | 0.009 | 0.006 |
| | (0.020) | (0.020) | (0.020) | (0.019) |
| Child age (month) | 0.002 | 0.002 | 0.002 | 0.002 |
| | (0.002) | (0.002) | (0.002) | (0.002) |
| Standardized BSID-III Cognitive Score | 0.014 | 0.013 | 0.013 | 0.012 |
| | (0.010) | (0.010) | (0.010) | (0.010) |
| Only child | -0.062* | -0.048 | -0.048 | -0.049* |
| | (0.024) | (0.025) | (0.025) | (0.024) |
| Grandparent is primary caregiver | 0.048 | 0.035 | 0.035 | 0.034 |
| | (0.025) | (0.026) | (0.026) | (0.025) |
| Primary caregiver has at least 9 yrs of schooling | -0.006 | -0.007 | -0.007 | -0.010 |
| | (0.031) | (0.031) | (0.031) | (0.031) |
| Primary caregiver has non-farm work | -0.005 | 0.008 | 0.008 | -0.004 |
| | (0.036) | (0.039) | (0.039) | (0.038) |
| Household asset index | 0.004 | -0.001 | -0.001 | 0.003 |
| | (0.015) | (0.013) | (0.013) | (0.012) |
| Father out-migrated | -0.002 | -0.009 | -0.009 | -0.007 |
| | (0.017) | (0.017) | (0.017) | (0.016) |
| Constant | 0.342*** | 0.266*** | 0.267*** | 0.202** |
| | (0.040) | (0.048) | (0.046) | (0.061) |
| Observations | 670 | 670 | 670 | 670 |
| R-squared | 0.37 | 0.42 | 0.42 | 0.43 |

In the regression, we control for village fixed effects. Standard errors in the parentheses are clustered at the village level.

* $p < 0.05$

** $p < 0.01$

*** $p < 0.001$.

social ties. Drawing on data from 819 children and their families enrolled in parenting centers in 50 villages in the Qinling mountain region of northwestern China, we describe overall participation rates, the geographic proximity of households to the ECD program, and the social ties of sample households. We also examine the correlations between social ties, geographic proximity, and participation in the community-based ECD program.

The results show that participation in the community-based ECD program was fairly high. On average, sample households attended one third (33.6%) of open days at the parenting center, which corresponds to about two days per week. This study does not include a comparison of participation rates across studies, as past studies of community-based ECD interventions vary in how participation is defined and measured. For example, some studies define participation as program enrollment or program completion rates [41], while others measure

participation by the number of hours a participant attended the program [18]. Additionally, some studies of community-based interventions do not report the full details of program implementation, such as the frequency of sessions during the program time span, and therefore do not report participation [42]. However, although we cannot compare to participation in other community-based ECD programs, logically, two days per week is fairly frequent for a caregiver to bring a child to an activity. Despite the relatively high overall participation rate, however, our results show that half (50.4%) of sample households attended fewer than 20% of the open days at the parenting centers, or less than one day per week on average, meaning that a large share of sample families did not take full advantage of the intervention.

We find that social interactions play an important role in program participation. After controlling for baseline covariates and geographic proximity, both the number of social ties and the average participation rate of a household's social ties are linked to increased program participation. This positive correlation between social ties and program participation is consistent with results from other studies of community-based programs, which similarly find that a greater number of social ties corresponds to greater program participation. For example, in a study of welfare participation in the United States, Bertrand, Luttmer, and Mullainathan (2000) [25] found that the quantity of contacts in a person's local area has a significant effect on participation in welfare programs.

However, we find that households had social ties with relatively few other households that were also eligible for the community-based ECD program. The average household had only five social ties to other households eligible to participate in the local parenting center, while villages had an average of 16 eligible households. This means that sample households interacted with less than a third of the eligible households within their local community, leaving potential to increase social ties. Considering that having one additional social tie is correlated with an increase in participation by 2.4 percentage points, increasing a household's social ties from five to 10, for example, would raise that household's participation rate from 34.2% to 46.2%.

Additionally, the participation rate of a household's social ties is correlated with an increase in individual participation in the ECD program, suggesting positive peer effects. This is also consistent with findings from non-ECD community-based programs [43–45], which have noted that individual decisions are affected by the choices of peers. The mechanism for peer effects varies, however. Some studies have found that social norms within the peer environment influence individual decisions [43], whereas others have found that information passed between peers is responsible for influencing decision-making [44]. In the context of our study, greater program participation within a household's social network may transmit information about the benefits of attending the parenting center, thusly raising individual participation.

When we examine geographic proximity to the parenting centers, we find that households, on average, live close to the parenting centers. The average distance from the parenting centers to sample households is about one km, and the majority of households (69%) live within one km from the center. This is a much shorter distance than reported in other (non-ECD) community-based programs [41, 46]. In a skill training program in rural Pakistan, for example, Cheema et al. (2019) [41] found that the average distance between households and the program center was six km, with considerable variability in distance.

However, despite the short distance between households and parenting centers in our study, geographic proximity remains a significant barrier to program participation. After controlling for both baseline covariates and all social network characteristics, an increase in distance from the parenting center by one km corresponds to a decrease in participation by seven percentage points. This finding is consistent with studies of non-ECD community-based programs [41, 46], which also find that increased distance is associated with lower rates of participation.

In the literature, the negative correlation between geographic proximity and program participation indicates that travel time and transportation costs of increased distance may outweigh the potential returns of the program [29, 46, 47]. In the context of our study, the costs of increased distance are associated with low mobility among caregivers, which may cause distance to be a significant barrier despite the fact that families live relatively close to the parenting centers. In our study, grandparents accounted for nearly half (45%) of visits to the parenting centers. However, compared to younger adults, elders are more likely to experience difficulties with mobility [48]. These mobility difficulties increase the time and cost of travel, and thus lower the likelihood of participation.

To empirically examine whether the low mobility of grandparent caregivers is contributing to the negative correlation between geographic proximity and program participation in our study, we conducted the same multivariate regression as Table 5 with an additional interaction term between "distance to the parenting" and "grandparent as the primary caregiver". The results are presented in S5 Table. In our most preferred specification (S5 Table, column 4), we find that distance serves as a much greater barrier for participation among grandparents. For all households, a one-kilometer increase in distance corresponds to a 6.2-percentage-point decrease in participation (row 4). For all households, one kilometer corresponds to a 6.2-percentage-point decrease in participation (row 4). For households in which a grandparent is the child's primary caregiver, however, a one-kilometer increase in distance corresponds to an additional decrease in participation by 7.8 percentage points (row 5), summing to a 15-percentage-point decrease in total (row 4 plus row 5). Therefore, for community-based ECD programs in rural China, geographic barriers are not only due to distance, but also to issues of low mobility among elderly caregivers, who make up a relatively large share of the primary caregivers in our sample. Considering these barriers, policies or programs that could lessen the physical burden of travel would likely have a significant positive effect on participation.

Taken together, our findings suggest that attention should be given to promoting social interactions among eligible households and eliminating geographic barriers in order to improve participation in community-based EDC programs. First, our results indicate that community-based ECD programs should incorporate activities to promote social interactions among eligible households. Extra bonding activities in the community-based program to promote social interaction among households may increase the number of social ties per household, which our data shows to be associated with increased program participation. The existence of peer effects also suggests that social ties may also amplify other efforts to increase participation. For example, efforts that increase the participation of other households in the village, such as providing public transportation, may lead to greater increases in individual participation due to peer effects. Second, geographic barriers can be reduced through public transportation, such as a shuttle bus that ferries residents living more than one km from the program location. This would not only reduce the travel and effort of walking; it would also mitigate the difficulty caused by variation in participant mobility, particularly that of grandparents.

This study makes two significant contributions to the literature. First, this paper is one of the first to quantitatively document program participation in a community-based ECD intervention, and it is the first study to do so in the context of rural China. Second, this is the first paper to examine factors that may predict participation in community-based ECD interventions. As community-based programs are increasingly promoted as the future of ECD programs in LMICs [10], this study provides empirical evidence that can be used to increase participation in community-based ECD programs in LMICs, which can improve overall program effectiveness and lead to greater long-term impacts on child development.

We also acknowledge two limitations of this study. First, our results on the links of geographic proximity and social ties to program participation in the community-based ECD

program are correlational, and we cannot identify causality. To identify the causal effect of geographic proximity and social interaction, we would need to randomize the location of the ECD parenting center and randomly form channels for social interactions, respectively. Second, as we mentioned in the methods section, we exclude long-term migrants from our sample. As a result, we do not know about the participation behavior of long-term migrants if the program were put forward at a large scale. Future studies should examine the effects of family out-migration on program participation and investigate ways to bring community-based programs to migrant populations.

## Supporting information

**S1 File.**
(DOCX)

**S2 File.**
(DOCX)

**S3 File.**
(DOCX)

**S4 File.**
(DOCX)

**S1 Fig. Participation rates and distance of social ties.** Source: Authors' Survey.
(TIF)

**S1 Appendix. Analysis of attrition.**
(DOCX)

**S1 Table. Correlates of attrition.**
(DOCX)

**S2 Table. Distribution of social ties with two alternative social tie definitions.**
(DOCX)

**S3 Table. Correlates of participation in the community-based ECD program (social ties defined as interacting at least once a month).**
(DOCX)

**S4 Table. Correlates of participation in the community-based ECD program (social ties defined as interacting at least twice a week).**
(DOCX)

**S5 Table. Correlates of participation in the community-based ECD program (adding distance*grandparents as main caregiver).**
(DOCX)

## Acknowledgments

We would like to thank the dedicated leaders and local cadres at the National Health Commission for their unparalleled assistance in implementing this study.

## Author Contributions

**Conceptualization:** Scott Rozelle.

**Formal analysis:** Yiwei Qian, Yi Ming Zheng.

**Funding acquisition:** Scott Rozelle.

**Methodology:** Yiwei Qian.

**Project administration:** Sarah-Eve Dill.

**Software:** Yi Ming Zheng.

**Writing – original draft:** Yiwei Qian, Yi Ming Zheng, Sarah-Eve Dill.

**Writing – review & editing:** Sarah-Eve Dill, Scott Rozelle.

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
