## [Decision Letter · Decision Letter 0]

23 Apr 2020

PONE-D-20-03940

Correlates of participation in community-based interventions: evidence from a parenting program in rural China

PLOS ONE

Dear Qian,

Thank you for submitting your manuscript to PLOS ONE. After careful consideration, we feel that it has merit but does not fully meet PLOS ONE’s publication criteria as it currently stands. Therefore, we invite you to submit a revised version of the manuscript that addresses the points raised during the review process.

Reviewer 1 has suggested some minor improvements in data and analysis that would be help improve the manuscript.

We would appreciate receiving your revised manuscript by Jun 07 2020 11:59PM. To enhance the reproducibility of your results, we recommend that if applicable you deposit your laboratory protocols in protocols.io, where a protocol can be assigned its own identifier (DOI) such that it can be cited independently in the future. For instructions see: http://journals.plos.org/plosone/s/submission-guidelines#loc-laboratory-protocols

We look forward to receiving your revised manuscript.

Kind regards,

Muhammad A. Z. Mughal

Academic Editor

PLOS ONE

Journal Requirements:

2.Please include additional information regarding the survey or questionnaire used in the study and ensure that you have provided sufficient details that others could replicate the analyses. For instance, if you developed a questionnaire as part of this study and it is not under a copyright more restrictive than CC-BY, please include a copy, in both the original language and English, as Supporting Information.

"SR received funding from the 111 Project (Grant no: B16031). The 111 project is funded by the Ministry of Education, China. The url to the funder information (http://www.moe.gov.cn/srcsite/A16/s7062/200608/t20060830_82287.html) is only in Chinese.

SR received funding from the National Natural Science Foundation of China (Grant no. 71703083). The url to the funder information can be found at http://www.nsfc.gov.cn/english/site_1/index.html;

SR received funding from the International Initiative for Impact Evaluation (3ie). The grant number in this case is not applicable. The url to the funder information can be found at https://www.3ieimpact.org;

SR received funding from the UBS Optimus Foundation. The grant number in this case is not applicable. The url to the funder information can be found at https://www.ubs.com/microsites/optimus-foundation/en.html;

SR received funding from the China Medical Board. The grant number in this case is not applicable. The url to the funder information can be found at https://chinamedicalboard.org;

SR received funding from the Bank of East Asia. The grant number in this case is not applicable. The url to the funder information can be found at https://www.hkbea.com/html/en/index.html;

SR received funding from the HBGDki Initiative. The HBGDki Initiative is funded by the Bill and Melinda Gates Foundation. The grant number in this case is not applicable. The url to the funder information can be found at https://www.gatesfoundation.org;

The funders had no role in study design, data collection and analysis, decision to publish, or preparation of the manuscript. "

Thank you for stating the following in the Competing Interests section:"The authors have declared that no competing interests exist."

We note that you received funding from a commercial source:"Bank of East Asia"

Reviewers' comments:

Reviewer's Responses to Questions

**Comments to the Author**

1. Is the manuscript technically sound, and do the data support the conclusions?

Reviewer #1: Yes

Reviewer #2: Yes

2. Has the statistical analysis been performed appropriately and rigorously? 

Reviewer #1: Yes

Reviewer #2: Yes

3. Have the authors made all data underlying the findings in their manuscript fully available?

Reviewer #1: Yes

Reviewer #2: No

4. Is the manuscript presented in an intelligible fashion and written in standard English?

Reviewer #1: Yes

Reviewer #2: Yes

5. Review Comments to the Author

Reviewer #1: Review: Correlates of participation in community-based interventions: evidence from a parenting program in rural China

Summary: The authors examine which characteristics predict participation in early childhood development programs in rural China. Proximity, social ties, peer participation are each significant predictors.

Recommendation:

Minor revision

General Notes:

• Providing parenting centers throughout rural China is a very interesting intervention and participation is an important component of understanding the overall effects and costs/benefits. The emphasis throughout the paper seems to be the effects of distance first, followed by social ties. I would recommend (though not require) reversing this ordering of emphasis. Focusing on the association between participation and distance is less novel because the relationship is self-evident.

• The authors make in interesting point in the discussion regarding the role of distance and the mobility of the caretakers: that geographic barriers are not due to distance alone but issues of low mobility. However, the authors do not provide evidence of this. Please add distance*grandparent caretaker to the models of participation in order to support the claim. If supported, this would add to the interest of the geographic point and perhaps justify its emphasis over the social ties.

• How was it determined that 3+ interactions = social tie? 3 seems like a high bar and households that interact 1 or 2 times per week would also seem to be socially tied. Sensitivity analyses should be performed around this threshold.

Specific Notes:

Line 180: Should be “our” not “on”

Check for additional typos throughout

Reviewer #2: The manuscript collected data of all children in the desired age range and their caregivers in randomly selected 100 village from 20 nationally-designated poverty counties. The statistical analysis been performed appropriately and rigorously，the conclusions has been drawn appropriately based on the data presented and statistical analysis.

The data are only available with some restrictions because of the involvement of human participants, data are available for researchers who meet the criteria for access to confidential data.

The manuscript presented in an intelligible fashion and written in very good English.

The research is valuable because this paper is one of the first to quantitatively document program participation in a community-based ECD intervention, and is the first paper to examine factors that may predict participation in community-based ECD interventions.

6. PLOS authors have the option to publish the peer review history of their article (what does this mean?). If published, this will include your full peer review and any attached files.

Reviewer #1: No

Reviewer #2: No

---

## [Author Response · Author response to Decision Letter 0]

8 Jul 2020

Response to Reviewer 1

General Comment 1: The authors examine which characteristics predict participation in early childhood development programs in rural China. Proximity, social ties, peer participation are each significant predictors.

Response to General Comment 1: 

Thank you for your thoughtful read.

Comment 1: Providing parenting centers throughout rural China is a very interesting intervention and participation is an important component of understanding the overall effects and costs/benefits. The emphasis throughout the paper seems to be the effects of distance first, followed by social ties. I would recommend (though not require) reversing this ordering of emphasis. Focusing on the association between participation and distance is less novel because the relationship is self-evident.

Response to Comment 1. 

We thank the reviewer for this suggestion. In the revised manuscript, we have reversed the emphasis to describe the role of social ties first. In doing so, we have made structural changes throughout the manuscript. All of these revisions were made with tracked changes in the manuscript for your review.

Additionally, we would like to note that we believe both distance and social ties are equally important correlates of participation to study in the context of a community-based early childhood development (ECD) program. To date, there is a general lack of quantitative evidence on the correlates of participation in community-based ECD programs. As such, our findings of both distance and social ties offer a contribution to the literature, particularly in a developing setting such as rural China, where there is a need for increased ECD investment. 

Comment 2: The authors make in interesting point in the discussion regarding the role of distance and the mobility of the caretakers: that geographic barriers are not due to distance alone but issues of low mobility. However, the authors do not provide evidence of this. Please add distance*grandparent caretaker to the models of participation in order to support the claim. If supported, this would add to the interest of the geographic point and perhaps justify its emphasis over the social ties.

Response to Comment 2. 

Thank you for your thoughtful suggestion. 

We have re-performed our correlational analysis with an additional interaction term between distance to the parenting center and grandparents as the primary caregiver. The results are presented in S5 Table (included at the end of this letter to facilitate your review). In our most preferred specification (S5 Table, column 4), we find that distance serves as much greater barrier for participation among grandparents compared. For all households, one kilometer corresponds to a 6.2-percentage-point decrease in participation (row 4). For households in which a grandparent is the child’s primary caregiver, however, a one-kilometer increase in distance corresponds to an additional decrease in participation by 7.8 percentage points (row 5), summing to a 15-percentage-point decrease in participation in total (row 4 plus row 5). This result supports our point that the geographic barriers are not due to distance alone, but also to the low mobility of elderly caregivers.

In the revised manuscript, we have added a discussion of S5 Table on Page 23:

“To empirically examine whether the low mobility of grandparent caregivers is contributing to the negative correlation between geographic proximity and program participation in our study, we conducted the same multivariate regression as Table 5 with an additional interaction term between “distance to the parenting center” and “grandparent as the primary caregiver.” The results are presented in S5 Table. In our most preferred specification (S5 Table, column 4), we find that distance serves as a much greater barrier for participation among grandparents. For all households, a one-kilometer increase in distance corresponds to a 6.2-percentage-point decrease in participation (row 4). For all households, one kilometer corresponds to a 6.2-percentage-point decrease in participation (row 4). For households in which a grandparent is the child’s primary caregiver, however, a one-kilometer increase in distance corresponds to an additional decrease in participation by 7.8 percentage points (row 5), summing to a 15-percentage-point decrease in total (row 4 plus row 5). Therefore, for community-based ECD programs in rural China, geographic barriers are not only due to distance, but also to issues of low mobility among elderly caregivers, who make up a relatively large share of the primary caregivers in our sample. Considering these barriers, policies or programs that could lessen the physical burden of travel would likely have a significant positive effect on participation.”

We also changed the covariate “mother is primary caregiver” to “grandparent is primary caregiver” in Table 5 and S1 Table for consistency in the analysis. The updated results in Table 5 and Table S1 are almost the same as that in Table 5 and S1 Table in the original manuscript. 

Please find the S5 Table in the revised manuscript or the response to the reviewer letter.

Comment 3: How was it determined that 3+ interactions = social tie? 3 seems like a high bar and households that interact 1 or 2 times per week would also seem to be socially tied. Sensitivity analyses should be performed around this threshold.

Response to Comment 3. 

Thank you for this note. We agree with the reviewer that households interacting 1 or 2 times per week might also be deemed “socially tied.” In fact, would like to apologize for a typo in the manuscript regarding our definition of a social tie. In our analysis, households that interacted at least once per week (rather than three times per week) are considered to have a social tie. We have corrected this typo in the revised manuscript, and we have done a careful check for any other typos.

In addition, we conducted a sensitivity analysis of the definition of social ties. In S2 Table, we first show the distribution of the number of social ties, with social ties defined as interacting at least once per month (less frequent than the original definition) and interacting at least twice per week (more frequent than the original definition), respectively. Next, in S3 Table and S4 Table, we present the correlates of parenting center participation rates. The results find consistent correlations between the number of social ties and participation in the parenting center, regardless of the definition for a social tie. Thus, in S3 Table and S5 Table, as in Table 5 of the original manuscript, the number of social ties is consistently and positively correlated with participation. 

When we examine the correlation between the average participation of social ties and individual participation, we find that when we define social ties as interacting at least twice per week (S4 Table), the correlations are significant and positive, consistent with our original Table 5. However, the correlation is not statistically significant when we use a looser definition of social ties (at least once a month; S3 Table), although the coefficient is still positive. These results demonstrate that our original definition of social ties (interacting at least once per week) is an adequate measure.

In the revised manuscript, we have added a short discussion of S2, S3 and S4 Tables to confirm the validity of our definition of social ties (page 19):

“To check the robustness of our findings, we performed the same multivariate regression analysis as Table 5, using two different definitions social ties. S2 Table shows the distribution of number of social ties in our sample, using two alternative definitions for social ties. Panel A shows the distribution when social ties are defined as interacting at least once per month (less frequent than the original definition), and Panel B presents the distribution when social ties are defined as interacting at least twice per week (more frequent than the original definition). S3 Table and S4 Table present the regression results with social ties defined as interacting at least once per month and defined as interacting at least twice per week, respectively. The results find consistent correlations between the number of social ties and program participation, regardless of the definition of a social tie. However, although correlation between the average participation rate of social ties and individual participation remains significant when we use the stricter definition of a social tie (at least twice a week; S4 Table), the correlation is not statistically significant when we use the looser definition of social ties (at least once a month; S3 Table). Geographic proximity remains significantly correlated to program participation at a similar magnitude in all tables.”

Please find the S2 Table, S3 Table and S5 Table in the revised manuscript or the response to the reviewer letter.

Response to Reviewer 2

General Comment 1: The manuscript collected data of all children in the desired age range and their caregivers in randomly selected 100 village from 20 nationally-designated poverty counties. The statistical analysis been performed appropriately and rigorously，the conclusions has been drawn appropriately based on the data presented and statistical analysis.

The data are only available with some restrictions because of the involvement of human participants, data are available for researchers who meet the criteria for access to confidential data.

The manuscript presented in an intelligible fashion and written in very good English.

The research is valuable because this paper is one of the first to quantitatively document program participation in a community-based ECD intervention, and is the first paper to examine factors that may predict participation in community-based ECD interventions.

Response to General Comment 1:

 Thank you for your thoughtful read of this paper and your kind words.

---

## [Decision Letter · Decision Letter 1]

26 Aug 2020

Correlates of participation in community-based interventions: evidence from a parenting program in rural China

PONE-D-20-03940R1

Dear Dr. Qian,

We’re pleased to inform you that your manuscript has been judged scientifically suitable for publication and will be formally accepted for publication once it meets all outstanding technical requirements.

Kind regards,

Muhammad A. Z. Mughal, PhD

Academic Editor

PLOS ONE

Additional Editor Comments (optional):

Reviewers' comments:

Reviewer's Responses to Questions

**Comments to the Author**

1. If the authors have adequately addressed your comments raised in a previous round of review and you feel that this manuscript is now acceptable for publication, you may indicate that here to bypass the “Comments to the Author” section, enter your conflict of interest statement in the “Confidential to Editor” section, and submit your "Accept" recommendation.

Reviewer #1: All comments have been addressed

2. Is the manuscript technically sound, and do the data support the conclusions?

Reviewer #1: Yes

3. Has the statistical analysis been performed appropriately and rigorously? 

Reviewer #1: Yes

4. Have the authors made all data underlying the findings in their manuscript fully available?

Reviewer #1: Yes

5. Is the manuscript presented in an intelligible fashion and written in standard English?

Reviewer #1: Yes

6. Review Comments to the Author

Reviewer #1: (No Response)

7. PLOS authors have the option to publish the peer review history of their article (what does this mean?). If published, this will include your full peer review and any attached files.

Reviewer #1: No

---

## [Editor Report · Acceptance letter]

28 Aug 2020

PONE-D-20-03940R1 

Correlates of participation in community-based interventions: evidence from a parenting program in rural China 

Dear Dr. Qian:

I'm pleased to inform you that your manuscript has been deemed suitable for publication in PLOS ONE. Congratulations! Your manuscript is now with our production department. 

Kind regards, 

on behalf of

Dr. Muhammad A. Z. Mughal 

Academic Editor

PLOS ONE